# The Phylogeography of Deciduous Tree *Ulmus macrocarpa* (Ulmaceae) in Northern China

**DOI:** 10.3390/plants13101334

**Published:** 2024-05-12

**Authors:** Hang Ye, Yiling Wang, Hengzhao Liu, Dingfan Lei, Haochen Li, Zhimei Gao, Xiaolong Feng, Mian Han, Qiyang Qie, Huijuan Zhou

**Affiliations:** 1Key Laboratory of Resource Biology and Biotechnology in Western China, Ministry of Education, College of Life Sciences, Northwest University, Xi’an 710069, China; phdye@stumail.nwu.edu.cn (H.Y.); hengzhaoliu@stumail.nwu.edu.cn (H.L.); leidingfan@stumail.nwu.edu.cn (D.L.); lihaochen@stumail.nwu.edu.cn (H.L.); gaozhimei@stumail.nwu.edu.cn (Z.G.); 2School of Life Sciences, Shanxi Normal University, Taiyuan 030031, China; 221112089@sxnu.edu.cn (X.F.); 222112039@sxnu.edu.cn (M.H.); 221112090@sxnu.edu.cn (Q.Q.); 3Xi’an Botanical Garden of Shaanxi Province (Institute of Botany of Shaanxi Province), Xi’an 710061, China

**Keywords:** *Ulmus macrocarpa*, phytogeography, climate oscillations, Northern China, refugia

## Abstract

Disentangling how climate oscillations and geographical events significantly influence plants’ genetic architecture and demographic history is a central topic in phytogeography. The deciduous ancient tree species *Ulmus macrocarpa* is primarily distributed throughout Northern China and has timber and horticultural value. In the current study, we studied the phylogenic architecture and demographical history of *U. macrocarpa* using chloroplast DNA with ecological niche modeling. The results indicated that the populations’ genetic differentiation coefficient (*N*_ST_) value was significantly greater than the haplotype frequency (*G*_ST_) (*p* < 0.05), suggesting that *U. macrocarpa* had a clear phylogeographical structure. Phylogenetic inference showed that the putative chloroplast haplotypes could be divided into three groups, in which the group Ⅰ was considered to be ancestral. Despite significant genetic differentiation among these groups, gene flow was detected. The common ancestor of all haplotypes was inferred to originate in the middle–late Miocene, followed by the haplotype overwhelming diversification that occurred in the Quaternary. Combined with demography pattern and ecological niche modeling, we speculated that the surrounding areas of Shanxi and Inner Mongolia were potential refugia for *U. macrocarpa* during the glacial period in Northern China. Our results illuminated the demography pattern of *U. macrocarpa* and provided clues and references for further population genetics investigations of precious tree species distributed in Northern China.

## 1. Introduction

Inferring the spatio-temporal diversity patterns and demography history of species has been central in phytogeography since its conception. Climatic-induced environmental alterations, coupled with geological phenomena, exert a significant influence on the geographical distribution of species and the dynamics of population demographics [1], especially in ancient tree species with relatively long evolutionary histories [2,3,4]. Being a vital vegetation type in the Northern Hemisphere, the origin and historical biogeography of temperate forest biome have received extensive attention since the nineteenth century [5,6,7]. Nevertheless, our comprehension of this subject remains constrained.

Climatic oscillations during Quaternary periods with more frequent glacial–interglacial cycles dramatically affected the species diversity pattern in Northern Hemisphere [8,9,10]. Within the realm of East Asia, it is posited that a minimum of four notable glaciations have taken place, presumably impacting the region’s flora and fauna, albeit the glacial progressions were less encompassing compared to Europe and North America [1,11,12]. Specifically, Northern China underwent drastic climatic fluctuations throughout the Quaternary period, yet it was never blanketed by extensive ice sheets [1,10,12]. The suitable habitat during the glacial period served as species’ refugia and played a critical role in species’ post-glaciation recolonization [13,14]. Previous research has proposed distinct refugia patterns for woody species in China, including single refugia, multiple refugia, microrefugia, and cryptic refugia [13,15,16]. However, a significant bulk of studies have been primarily fixated on the populations or species in Southern, Southwestern, and Northwestern China [9,17,18], leaving the northern zones under-researched and less explored.

The Elm family (Ulmaceae) is a moderately sized family consisting of approximately 50–60 species in the order Rosales. In Ulmaceae, the genera *Ulmus*, *Hemiptelea*, and *Planera* are predominantly deciduous and widely distributed in the northern temperate zone. Hence, the exploration of Ulmaceae’s biogeographic and demographic history could provide insights into the assembly of the temperate forest biome [19,20]. The previous study also emphasized the significance of Ulmaceae as a model system for investigating the boreal tree species’ historical biogeography and diversification patterns [21]. It has been reported that Ulmaceae emerged in the Early Cretaceous and could be separated into a temperate clade and a tropical clade [20,21]. Coalescent simulation suggested an East Asian origin of the temperate Ulmaceae clade during the Paleocene, consistent with the fossil records [21]. The macroclimatic niche of Ulmaceae species varies between the tropical and temperate clades. Ecological preferences play a crucial role in determining the geographical distribution of Ulmaceae plants [20]. Elm species are adaptable and resist drought, cold, and saline–alkali stresses. Comparative transcriptome analysis showed that positively selective genes may play essential roles in adapting to environmental changes for elms [22].

Serving as the diversity center, China has the highest Ulmaceae diversity worldwide [20]. *Ulmus macrocarpa*, mainly distributed across Northern China, is a precious woody germplasm resource and a key part of forest communities [23]. This tree can reach 20 m in height with a diameter of 40 cm at breast height. Its branches are light yellow–brown or pale reddish–brown; leaves are broad obovate or elliptic–obovate, 5–9 cm long, with a pointed tip. Winged fruits, 2.5–3.5 cm long, are hairy and have seeds in the middle. Flowering is in April–May; fruiting is in May–June, and leaves fall in October [24].

*U. macrocarpa* has important pharmaceutical, horticultural, and timber values [25,26,27]. Its wood is high-quality and commonly used in various applications. The seeds of *U. macrocarpa* have top-ranked weight, oil content, and decanoic acid among *Ulmus* species [24]. Its seed oil is utilized in food, industries, brewing, soy sauce production, and medicine. The bark also contains polysaccharides and mucilage for medicinal use. Additionally, the *U. macrocarpa* is a heliophyte with strong sprouting ability, capable of resisting cold and drought conditions.

As an important germplasm resource in Ulmaceae, although *U. macrocarpa* had significant economic value and high ecological adaptability, the advanced molecular phytogeography research on *U. macrocarpa* received less attention compared to other Ulmaceae species. A previous study observed high phenotypic variations in *U. macrocarpa* populations. The micromorphological features of the leaf epidermis in *U. macrocarpa* were influenced by environmental factors such as temperature and altitude [24]. However, the genetic diversity and differentiation of *U. macrocarpa* was still unknown. Furthermore, owing to a lack of nuclear genome data, such research on elms is singularly lagging behind other species, with existing research limited to morphology, allozyme markers, molecular markers, and transcriptomes [10,22,28,29,30,31,32,33,34,35,36]. The chloroplast genome was popularly utilized in plant population genetics, polymorphism investigations, ecological and evolutionary studies, and DNA barcoding owing to its characteristics of small genome size, relatively slow evolutionary rates, and maternal inheritance. Meanwhile, the application of chloroplast DNA contributed a lot to plant phytogeography. Although the chloroplast genome was relatively conserved, single nucleotide polymorphisms (SNPs) frequently occurred, providing valuable variation resources for intra- or interspecific phytogeography research.

Thanks to the rapid development of sequencing technology, the complete chloroplast genome of *U. macrocarpa* has been published [21], providing an excellent opportunity for genetic investigation. The *ycf1* was the most promising plastid DNA barcode of land plants, which was more variable than other chloroplast barcodes such as *matK* and *rbcL* [37,38,39]. So far, the chloroplast *ycf1* has been widely used in phylogenic and phytogeographic studies [40,41,42,43,44]. In this study, in addition to *ycf1*, three chloroplast intergenic spacer regions *rpl32–trnL* [45], *atpB–rbcL* [46], and *trnH–psbA* [47] were also used to investigate phytogeography of *U. macrocarpa* because of their stable amplification success and high variability. This study aimed to achieve the following: (1) reveal the genetic diversity and genetic differentiation landscape of *U. macrocarpa* populations; (2) unveil the impact of climate heterogeneity and geographical isolation on *U. macrocarpa* population genetic architecture; (3) explore the demographic history of *U. macrocarpa*; and (4) discuss its refugia pattern during the glacial period. Our results would illuminate the demographic history of valuable tree species *U. macrocarpa* and offer a theoretical foundation for further biodiversity investigation of the temperate forest biome in Northern China and even East Asia.

## 2. Materials and Methods

### 2.1. Sample Collection, DNA Extraction, and Sequencing

We sampled 110 individuals from 22 populations (5 individuals per population) throughout the geographical distribution of *U. macrocarpa* in Northern China (Figure 1; Appendix A). All samples from each population were separated by a distance exceeding 20 m, and fresh leaves were gathered for DNA extraction. The genome DNA was extracted from fresh leaves using a modified CTAB method [48]. The primers of *ycf1*, *rpl32–trnL*, *atpB-rbcL*, and *trnH-psbA* (Appendix A) were designed according to the complete chloroplast genome of *U. macrocarpa* (NCBI accession number: MT165937.1) [21] using Primer Premier software 5 [49]. The DNA quality was then assessed through ultraviolet spectrophotometry and 0.8% agarose gel electrophoresis, followed by storage at −20 °C for subsequent use.

The PCR protocols were adapted from Liu et al. [36], with the reaction carried out in a 25 μL volume comprising 4 μL of template DNA (30 ng/μL), 1 μL of primers, 12.5 μL of 2× MasterMix, and 7.5 μL of ddH_2_O. The PCR process included pre-denaturation (94 °C for 5 min), denaturation (94 °C for 1 min), annealing (55 °C for 1 min), elongation (72 °C for 1.5 min) for 35 cycles, and final elongation (72 °C for 10 min). The PCR products were detected using 2% agarose gel electrophoresis and confirmed through automated gel imaging analysis before being sent to Sangon Biotech (Shanghai, China) for sequencing.

### 2.2. Haplotype Identification and Genetic Diversity Analysis

After sequencing, the chloroplast DNA fragments were aligned using MAFFT [50] and then pruned by GBLOCKS [51]. Using the DNASP 6 software [52], the chloroplast haplotype was identified, followed by the calculation of nucleotide diversity (π) and haplotype diversity (H*d*). The PERMUT 2.0 program [53] was used to estimate the genetic parameters, including gene diversity (*h*_S_), total gene diversity (*h*_T_), geographical total haplotype diversity (*V*_T_), and geographical average haplotype diversity (*V*_S_). The presence of phylogeographic structure was assessed by testing whether *N*_ST_ (genetic differentiation takes into account distances among haplotypes) was significantly higher than *G*_ST_ (population differentiation that does not consider distances among haplotypes) with 1000 times permutation tests. To quantify and assess the distribution of genetic variation within and between populations, a hierarchical analysis of molecular variance (AMOVA) was implemented by using ARLEQUIN 3.5 software [54] with 1000 times permutations.

### 2.3. Phylogeny Inference and Genetic Structure Estimation

Different methods were employed to disentangle the genetic structure and phylogenetic relationship of *U. macrocarpa* populations. Nei’s genetic distance was employed to reconstruct a neighbor-joining (NJ) tree using POPTREE 2 [55] with 1000 times bootstrap replicates. A principal component analysis (PCA) of all studied populations was performed using GENALEX 6.5 [56].

Bayesian clustering analysis was executed using STRUCTURE 2.3 software [57]. The posterior probability of grouping number (*K* = 1–10) was estimated by conducting 10 independent runs using 5 × 10^5^-step Markov chain Monte Carlo (MCMC) replicates, preceded by a 1 × 10^6^-step burn-in for each run to assess consistency. The optimal grouping number was determined using Δ*K* method as calculated in STRUCTURE HARVESTER [58]. Subsequently, the results from these 10 runs were aligned and consolidated with CLUMPP 1.1.2 [59], and the visualization of the results was generated using DISTRUCT 1.1 [60]. When considering the sampling location, the genetic structure was assessed through a spatial analysis of molecular variance using SAMOVA 2.0 [61]. Similar to the STRUCTURE analysis, the number of groups (*K*) of geographically adjacent populations was set from 1 to 10 in SAMOVA. The geographical distance was employed as the prior condition for the population’s genetic grouping, and the permutations test was set to 1000 times. The haplotype network was also constructed to reveal the phylogenetic relationship of putative haplotypes using POPART 1.7 [62].

To investigate the influence of isolation by geography and climate on the genetic differentiation of *U. macrocarpa* populations, the Mantel test and generalized linear model (GLM) were performed using the R package vegan [63]. Therein, pairwise FST values calculated from aligned sequences were utilized as a genetic distances matrix. The geographic distances matrix between populations was determined using GENALEX 6.5 software [56], and the environmental distances matrix was generated using PASSAGE 2 software [64].

### 2.4. Phylogeographical Pattern and Demographic History Inference

The BEAST 1.8.4 software [65] was used to estimate the haplotype divergence time of *U. macrocarpa*, in which *U. pulima* was employed as the outgroup. JMODELTEST 2 [66] was adopted to estimate the optimum substitution model according to the Akaike Information Criterion [67], and T92 was finally confirmed as the best-fit model. The evolutionary rates were set as 1.1 to 2.9 × 10^−9^ mutations per site per year, according to a previous study [68]. The fossil record for *U. macrocarpa* (20.44–15.97 Ma, https://paleobiodb.org/classic/basicTaxonInfo?taxon_no=445605 (accessed on 25 February 2024)) was used to time tree calibration [69]. With the relaxed clock model in lognormal distribution, the MCMC process was repeated 5 × 10^7^ times by sampling every 20,000 generations to guarantee an effective sample size of over 200. We constructed a consensus tree with a posterior probability threshold exceeding 0.5 by utilizing TREE ANNOTATOR, following the exclusion of the initial 25% of trees as burn-in. The consensus tree was edited and visualized in FIGTREE.

To assess whether the species had experienced significant expansion, we used ARLEQUIN [54] to calculate the Tajima’s D [70] and Fu’s FS [71] values. Moreover, the sum of square deviation (SSD) and raggedness index (Rag) was also calculated in the mismatch distribution analysis (MDA) using ARLEQUIN. Phytogeography dynamics of *U. macrocarpa* were inferred via the Bayesian Binary MCMC (BBM) Method by using RASP 4 software [72]. Furthermore, the historical and contemporary gene flows during *U. macrocarpa* evolution were estimated by MIGRATE-N 3.7.2 [73] and BAYESASS 3 [74], respectively.

### 2.5. Ecological Niche Modeling

The maximum entropy modeling technique (MAXENT) [75] was utilized to predict the potential distribution of *U. macrocarpa* in China in different periods, including the contemporary period, mid-Holocene (MH, ~6 ka BP), Last Glacial Maximum (LGM, ~21 ka BP), and Last Interglacial (LIG, ~120–140 ka BP). In addition to the sampling sites in our study, the occurrence recodes of *U. macrocarpa* were also obtained from the Chinese Virtual Herbarium (CVH, http://www.cvh.ac.cn/ (accessed on 25 February 2024)) and the National Specimen Information Infrastructure (NSII, http://www.nsii.org.cn). After filtering, 59 records were finally retained for ecological niche modeling (ENM).

A total of 19 biological climate variables (bio1–bio19) at 2.5 arc-min resolution were downloaded from the WorldClim database (http://www.worldclim.org (accessed on 25 February 2024)). To avoid overfitting caused by the multicollinearity of climate variables, the VIF (Variance Inflation Factor) method was employed for variable selection using the R package usdm [76]. Climate variables with VIF > 20 and high correlation (|r| > 0.85) were removed [77]. After filtering, five climate variables were retained, namely, bio3, bio6, bio8, bio14, and bio16. The receiver operating characteristic (ROC) curve was used to test the accuracy of the MAXENT prediction. The fitness of suitable habitats was manually divided into a gradient classification corresponding to unsuitable (0–0.1), lowly suitable (0.1–0.25), moderately suitable (0.25–0.5), and highly suitable regions (>0.5), respectively.

To investigate the overlaps and differentiation of the niches between different periods of *U. macrocarpa* populations occupied, we computed Schoener’s *D* [78] and Hellinger distance *I* [79] using the ENMTOOLS [80]. The values of the two parameters varied from 0 (complete differentiation) to 1 (complete overlapping), which could be regarded as indicators for assessing the similarity of niches.

## 3. Results

### 3.1. Genetic Diversity and Differentiation

After alignment and manual correction, the concatenate chloroplast sequence was 1668 bp in length, including 41 polymorphic sites. A total of eighteen haplotypes were identified among 110 accessions (Figure 1; Appendix A). The haplotypes H8 and H4 were the most widely distributed, shared by nine and eight populations, respectively, followed by H13 (shared by six populations). In contrast, the haplotypes H5, H6, H10, H12, H16, and H17 occurred in only single population. Most populations possessed multiple haplotypes, while only one haplotype (H8) was found in the SHHLC.

We observed the high genetic diversity of *U. macrocarpa* populations. The total haplotype diversity (*Hd*) and nucleotide diversity (π) were 0.901 and 0.0053, respectively. Furthermore, the genetic parameters *h*_T_, *h*_S_, *V*_T_, and *V*_S_ were 0.889, 0.397, 0.893, and 0.292, respectively. For different populations, the haplotype diversity ranged from 0 to 0.9. The population YQS presented the highest nucleotide diversity, corresponding to 0.0072, followed by the population QS with the nucleotide diversity of 0.0063. However, the nucleotide diversity of the population SHHLC was 0, indicating that no genetic variation was detected within this population (Appendix A).

The NJ tree (Figure 2a), PCA (Figure 2b), and STRUCTURE analysis (Figure 2c) consistently divided these populations into three groups. In the structure analysis, the Δ*K* had the highest value when *K* = 3, followed by *K* = 2 (Appendix A), indicating that the best grouping pattern was three among these populations. Moreover, the SAMOVA showed a higher *F*_CT_ value (0.5320) when *K* = 3 than that (0.48322) when *K* = 2, suggesting that the best geographical grouping was also three. As expected, we observed that three groups had a strong geographic basis (in group Ⅰ, the population was mainly located in Shanxi, including KL, HCS, ZJS, WTS, QLY, HLMKZ, YQS, DGH, and MHS; in group Ⅱ, the population mainly located in Hebei and Beijing, including DQG, DHS, HLS, QFS, XWTS, BHS, and SFS; in group Ⅲ, the population was mainly located in Heilongjiang, Jilin, and Liaoning, including QS, SDLC, HXLC, MDF, JH, and SHHLC) (Figure 1 and Figure 2; Appendix A). The haplotype network presented a consistent geographic distribution pattern with the STRUCTURE and SAMOVA, in which the haplotypes originating from the same group had a closer relationship (Figure 1b). The genetic differentiation between populations was significantly higher when computed using a distance matrix than when using haplotype frequencies (*N*_ST_ (0.518) > *G*_ST_ (0.240), *p* < 0.05), indicating a significant phylogeographic structure within *U. macrocarpa*.

Furthermore, we investigated the genetic differentiation of these three *U. macrocarpa* phytogeographical groups in detail. The hierarchical AMOVA revealed that more than half the amount of variation (53%) occurred between three groups, and only 7% presented differences among populations within groups, whereas 40% of the variation was within populations (Table 1). The genetic differentiation between group Ⅰ and group Ⅲ was relatively high (*F*_CT_ = 0.587), followed by group Ⅱ and group Ⅲ (*F*_CT_ = 0.554) and group Ⅰ and group Ⅱ (*F*_CT_ = 0.440) (Table 1). We also found that the genetic differentiation among *U. macrocarpa* populations was significantly associated with geographical and climate isolation. Mantel tests and GLM (Figure 3) unveiled a significant correlation between genetic and geographic distance (r = 0.5137, R^2^ = 0.2639, F = 82.08, *p* = 0.001) and between genetic and climatic distance (r = 0.1841, R^2^ = 0.03389, F = 8.031, *p* = 0.005).

### 3.2. Phytogeographical Pattern and Demographic History

Demographic histories played an important role in shaping the current population’s diversity pattern. The Bayesian Inference (BI) tree constructed by BEAST suggested that 18 putative haplotypes could also be divided into three groups (Figure 4), which was consistent with the haplotype network (Figure 1b) and population structure analysis (Figure 2). Group Ⅰ consisted of ten haplotypes (H1, H2, H3, H4, H5, H6, H7, H9, H10, and H17), while groups Ⅱ and Ⅲ contained five (H12, H13, H14, H15, and H16) and three (H8, H11, and H18) haplotypes, respectively.

The differentiation time of *U. macrocarpa* with the outgroup *U. pulima* was ~20.7 Ma (95% HPD: 15.9–28.2 Ma) in the middle Miocene. The divergence time of the most recent common ancestor of all *U. macrocarpa* haplotypes was estimated at ~17.1 Ma (95% HPD: 5.17–13.33 Ma). Subsequently, the divergence between group Ⅰ and Ⅱ occurred at ~10.4 Ma (95% HPD: 6.6–14.2 Ma). We observed that the diversification within each group all occurred in the late Miocene. Specifically, the divergence time within groups Ⅰ, Ⅱ, and Ⅲ were estimated at 6.48 Ma, 6.79 Ma, and 7.02 Ma, respectively. Overall, the intragroup divergence continuously occurred from the later Miocene to Pliocene, where the approximate more recent differentiation time of the intragroup haplotypes for *U. macrocarpa* was during the Quaternary Era (e.g., H6 and H17, H12 and H16). According to the geographical distribution (Figure 1a) and genetic structure (Figure 1b and Figure 2), all putative haplotypes were assigned to three states (labeled A–C) in RASP analysis (Figure 4). Akin to the results of divergence time estimation, state A was inferred as the relatively ancestral state of *U. macrocarpa*. Nine dispersal and four vicariance events were identified during the *U. macrocarpa* evolution. The common ancestral of *U. macrocarpa* experienced dispersal events to form state C and the common ancestral of states A and B. Subsequently, vicariance and dispersal events were simultaneously defined as the splitting of an ancestral lineage into two descendant lineages (states A and B) that were divided between two adjacent regions.

Although the *U. macrocarpa* population, as well as the putative haplotypes, could be relatively divided into three phytogeographical groups (Figure 1, Figure 2 and Figure 4) with significant genetic differentiation (Table 1; Figure 3), we observed some stable admixtures among different groups (Figure 2c). Hence, we performed the MIGRATE and BAYESSAS analysis to explore the gene introgression between different groups (Table 2). The historical and contemporary gene flow ranged from 0.6250 to 1.3744 and 0.0068 to 0.0420, respectively. The historical and contemporary gene flows were relatively high between groups Ⅰ and Ⅲ.

### 3.3. Ecological Niche Modeling and Population Size Changing

Mismatch distributions analysis showed that the species experienced a recent expansion, as supported by uniformly non-significant SSD (0.0036, *p* > 0.05) and HRag (0.0154, *p* > 0.05) values, although Tajima’s D (2.6327, *p* > 0.05) and Fu’s FS (−2.7498, *p* > 0.05) values were not significant in neutrality test. To further explore the population dynamics of *U. macrocarpa*, ecological niche modeling was performed using the MAXENT method.

After filtering, six reserved climate variables (bio2, bio7, bio8, bio12, bio14, and bio15) had a dominant influence (percentage contribution = 87.6%) on the geographic distribution of *U. macrocarpa*. Furthermore, the high AUC values (>0.900) indicated that all MAXENT models exhibited high predictive ability.

The predictions of the current distribution for *U. macrocarpa* fit relatively well with its actual distribution (Figure 5) in Northern China, including Shaanxi, Shanxi, Shandong, Hebei, and Liaoning. During the LGM, the potential distribution of *U. macrocarpa* was smaller than that at present. The relatively lower *D* (0.7229) and *I* (0.9306) values indicated a significant niche differentiation between the LGM and the current period (Table 3). During the LIG, the potential range of this species was more extensive than that of the current period, with a northern expansion. The *U. macrocarpa* occupied more suitable northerly habitat areas such as Inner Mongolia in this period.

## 4. Discussion

### 4.1. High Genetic Diversity and Significant Genetic Differentiation of U. macrocarpa

Genetic diversity serves as the foundational material essential for biological evolution and represents the predominant origin of biodiversity [81]. High genetic diversity for species represents a successful adaptive strategy for coping with diverse habitats and environmental conditions, thereby enhancing the ability of plants or populations to access and establish in novel habitats [82]. In this study, high levels of genetic diversity were observed in *U. macrocarpa* populations (Appendix A), consistent with the rich phenotypic diversity of this species in previous studies [24]. The total gene diversity of *U. macrocarpa* (*h*_T_ = 0.889) was higher than the mean total gene diversity (*h*_T_ = 0.67) detected in 170 plant species, of which chloroplast DNA markers have been used [83]. The observed high genetic diversity of *U. macrocarpa* populations is likely attributable to the gradual accumulation of nucleotide mutations over extended evolutionary periods. Generally, widespread species tend to exhibit greater genetic variability compared to those with limited distributions. [84]. As a dominant tree species in forest communities, *U. macrocarpa* is widely distributed across Northern China, leading to high genetic diversity.

A significant phylogeographic structure (*N*_ST_ (0.518) > *G*_ST_ (0.240), *p* < 0.05) was found in *U. macrocarpa*. Subsequently, the clustering analysis (Figure 2) and SMOVA consistently divided *U. macrocarpa* populations into three groups with significant genetic differentiation (Table 1). Numerous factors influence the genetic differentiation among plant populations, with geographical isolation and climatic heterogeneity identified as pivotal drivers [84]. Spatial heterogeneity of the environment exerts differential selection pressures on natural populations, potentially leading to the rapid differentiation of populations [85,86]. Our study found significant Isolation-by-Distance and Isolation-by-Environment patterns via the Mantel test (Figure 3). The topography in Northern China is highly diversified. The uplifts of mountains such as Taihang, Yanshan, and Yinshan in Northern China have served as practical barriers to prevent dispersal and introgression, thus promoting genetic diversification of species. With geographical isolation, the split *U. macrocarpa* populations gradually adapted to local climate conditions. Additionally, *U. macrocarpa* is dioecious and wind-pollinated, limiting pollen dispersal to short distances. Consequently, the diminished gene flow (Table 2) due to habitat fragmentation has resulted in pronounced genetic divergence among *U. macrocarpa* populations in Northern China.

### 4.2. Demographic History Inference of U. macrocarpa

The populations of *U. macrocarpa* had a significant phylogeographic pattern (Figure 1 and Figure 2). The mismatch distributions analysis of the chloroplast sequence indicated that the *U. macrocarpa* populations experienced a recent expansion (SSD = 0.0036, *p* > 0.05; HRag = 0.0154, *p* > 0.05), which was consistent with the repaid diversification of *U. macrocarpa* during late Miocene to Quaternary (Figure 4). However, in previous studies, researchers suggested that the *U. macrocarpa*’s close relatives, *U. lamellosa,* did not undergo a significant expansion due to the limitation of uplift of mountains and complex geological conditions in Northern China [10,36]. We speculated that the high ecological adaptability of *U. macrocarpa* was the primary reason for its expansion to a broad range and the occupying of multiple gradient niches (Figure 1 and Figure 5) [24,25]. Additionally, the absence of a continuous ice sheet in Northern China in the Quaternary allowed for population expansion during the post-glacial recolonization and dispersal (Figure 5).

Based on the fossil calibration, although the intraspecific divergence of all *U. macrocarpa* haplotypes (17.1 Ma, 95% HPD: 14.6–23.8 Ma) most probably began during the early stage of middle Miocene (Figure 4), the blooming diversification within three *U. macrocarpa* phytogeographic groups occurred mainly during the late Miocene to Pliocene. This period is an important differentiation epoch for tree species in China. Evidence supporting late Miocene and Pliocene diversification of other temperate tree species has also been documented in many phytogeographic studies, such as *Acer* [87], *Cercidiphyllum* [88], and *Juglans* [2,89]. The uplift of the Tibetan–Himalayan Plateau in the late Miocene is seen as a key factor in triggering the diversification of plants in China due to changes in the Asian monsoon circulation, aridification in Central Asia, and China’s complex topography (~10–7 Ma) [90,91,92]. Northern China was impacted by both global climate change and the spread of aridity from West to East [93]. We speculated that the diversification of *U. macrocarpa* haplotypes was associated with the monsoon circulation and aridification in Northern China during this period.

Additionally, the deformation of mountains located in Northern China, such as the Taihang, Yanshan, Yinshan, and Changbai Mountains, simultaneously occurred with the significant uplift of the Tibetan–Himalayan Plateau [94,95]. In addition to severing as geographical barriers, these mountains sometimes provided dispersal corridors, allowing for range expansions and population connectivity [96,97]. For example, during the Miocene, the uplift height of the northern part of the Taihang Mountains was merely half that of the western zone of the Yanshan Mountains [98]. The relatively modest elevation of the Taihang Mountains did not pose as a geographical barrier, thus failing to obstruct the migration of *U. macrocarpa*, which was consistent with the results of the RASP analysis (Figure 4). The *U. macrocarpa* ancestral group Ⅰ, mainly distributed in Shanxi and Henan provinces (Appendix A; Figure 1), crossed the Taihang Mountains corridor and spread to Hebei and further northeast regions such as Heilongjiang and Jilin provinces, leading to the global diversification landscape of *U. macrocarpa* in Northern China.

### 4.3. Potential Refugia under the Climate Oscillation in Quaternary

The Quaternary glacial–interglacial cycles had significant impacts on the differentiation and migration patterns of species [99]. The onset of certain haplotype divergence was more recent (Figure 4), implying that historical processes and climatic oscillations during the Quaternary period have also shaped the present-day distribution of *U. macrocarpa*. Akin to other temperate tree species, Quaternary climatic oscillations and associated environmental changes may have made the diversification within *U. macrocarpa* in Northern China relatively complicated [2,10,36,84,89], which was also supported by the ecological niche modeling (Figure 5).

The climatic oscillations resulting from multiple glaciations have led to shifts in the distribution of *U. macrocarpa* and the isolation of populations, potentially resulting in the development of distinct population structures characterized as “southern richness” and “northern purity” [10]. As expected, during the LGM, the *U. macrocarpa* distribution, showing a southward trend, was significantly smaller than those of the present day, as well as the LIG (Table 3; Figure 5), which was consistent with the previous study [36]. Furthermore, multiple geographically isolated refugia existed for forests across East Asia in LGM, which promoted intraspecific divergence and led to their speciation and diversification [14,99]. Under the glaciations, refugia are ideal habitats for conserving genetic diversity and generating unique alleles. Therefore, geographic areas displaying increased levels of genetic diversity are, thus, good candidates in the search for past refugia [100].

In this study, we speculated that the surrounding areas of Shanxi and Inner Mongolia, where the group Ⅰ mainly located, might be the major glacial refugia for *U. macrocarpa*. In addition to being the relatively ancestral group (Figure 4), it also possessed a higher haplotype and genetic diversity (*Hd* = 0.8666, π = 0.00368) than that of groups Ⅱ (*Hd* = 0.7176, π = 0.00210) and Ⅲ (*Hd* = 0.58391, π = 0.00295). The ecological niche modeling showed a highly suitable level, especially in Shanxi, for *U. macrocarpa* during the LGM (Figure 5). Interestingly, we found that haplotype 4 was the most dominant in group Ⅰ (Figure 1; Appendix A), occurring in eight populations (seven in Shanxi and one in Inner Mongolia) and located at the center of group Ⅰ in the haplotype network. Furthermore, the populations with haplotype 4, such as YQS and HLMKZ, had higher haplotype and genetic diversity (Appendix A). These results further indicated the reliability of refugia.

We also observed that haplotypes 13 and 8 were widely distributed in groups Ⅱ and Ⅲ, respectively (Figure 1; Appendix A). Each phylogeographical group contained its unique dominant haplotype with high differentiation from each other (Figure 1 and Figure 2; Table 1 and Appendix A), indicating a parallel evolution pattern of *U. macrocarpa* populations in Northern China. The Taihang and Yanshan Mountains were uplifted sharply along with the Himalayan orogeny movement to finally form in the Quaternary [84], which dramatically resisted the recent gene flow between different groups (Table 2), thus leading to the parallel evolution pattern within each group of *U. macrocarpa*.

## 5. Conclusions

In this study, using chloroplast DNA, we analyzed the phytogeography and demographical history of the temperate tree species *U. macrocarpa*. The results showed that *U. macrocarpa* had a significant phylogeographical structure with high genetic diversity. A total of 18 putative haplotypes were divided into three groups. The divergence of intraspecies haplotypes occurred during the Miocene–Pliocene, which was associated with major Tertiary geological and/or climatic events in Northern China. Quaternary climatic oscillations also significantly impacted the distribution of *U. macrocarpa*. The surrounding areas of Shanxi and Inner Mongolia might be regarded as refugia for *U. macrocarpa* in the glacial period. The uplift of mountains in the Quaternary resisted the gene flow between different phylogeographical groups, leading to a potential parallel evolution pattern of *U. macrocarpa* populations in Northern China. Our results broaden our understanding of the demographical history and genetic architecture of forests in Northern China.

## Figures and Tables

**Figure 1 plants-13-01334-f001:**
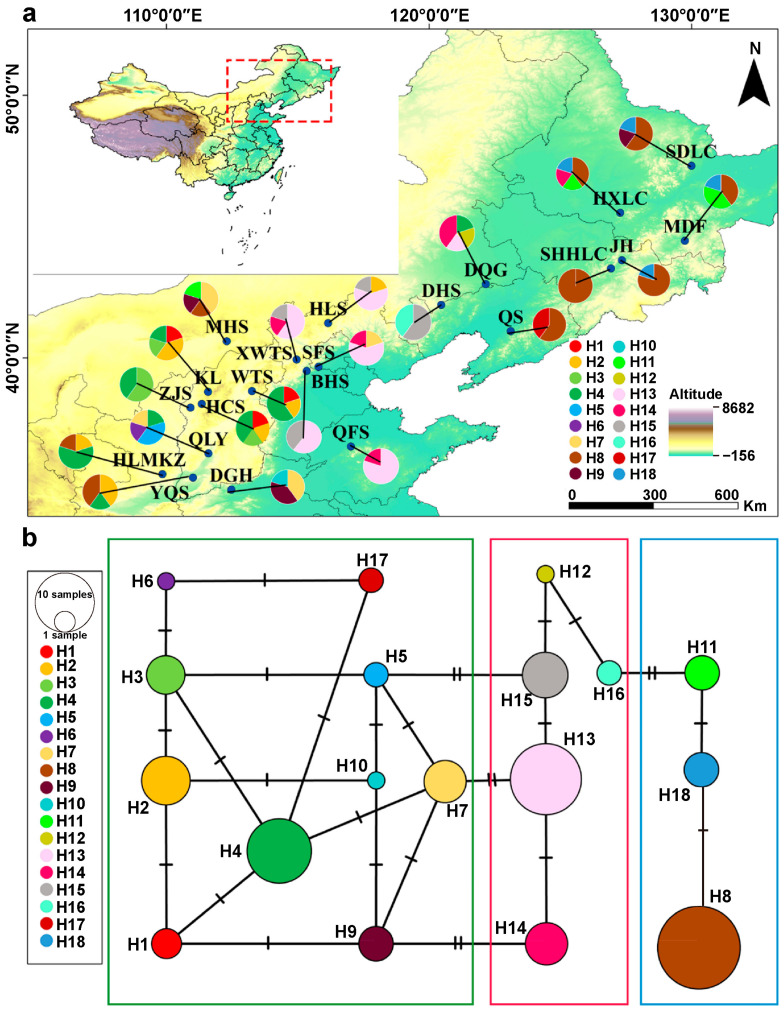
Sampling locations, haplotype distributions (**a**), and the network (**b**) of *U. macrocarpa* in Northern China. The pie chart represents the proportion of haplotypes in each population. Three frames in different colors (green, red, and blue) in network (**b**) correspond to the three different haplotype phylogenetic groups.

**Figure 2 plants-13-01334-f002:**
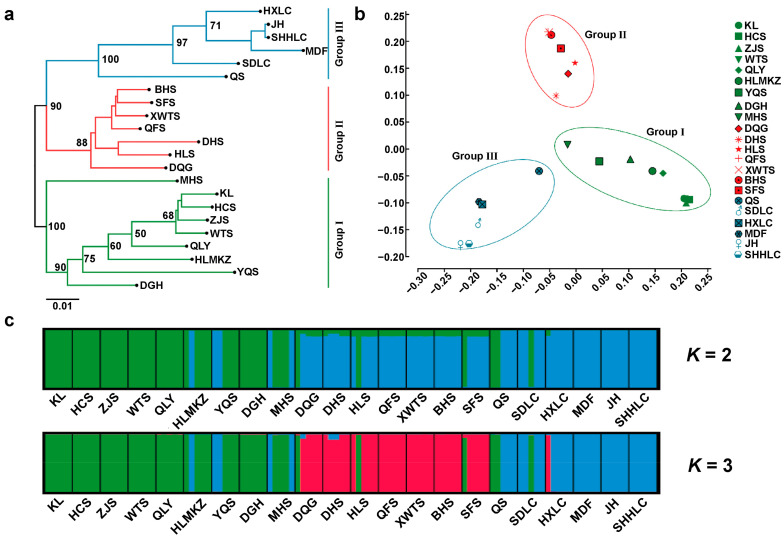
The genetic structure and phylogenic relationship of *U. macrocarpa* populations. (**a**) The population NJ tree of *U. macrocarpa*. (**b**) The PCA of *U. macrocarpa* populations. (**c**) The genetic structure for *K* = 2 and *K* = 3 for *U. macrocarpa* populations.

**Figure 3 plants-13-01334-f003:**
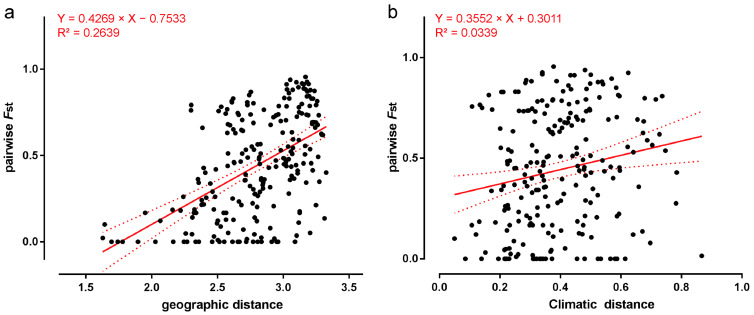
The distribution of pairwise *F*_ST_ along the geographic distance (**a**) and climatic distance (**b**), respectively. The red line represents the generalized linear model (GLM) equation (solid line) with 95% confidence interval (dotted line). (**a**) Genetic distance was positively correlated with geographic distance. GLM: Slope = 0.4269 (95% CI: 0.334–0.5198); Y-intercept = −0.7533 (95% CI: −1.015–−0.4917); X-intercept = 1.764 (95% CI: 1.465–1.961); R^2^ = 0.2639; F = 82.08; *p* = 0.001. (**b**) Genetic distance was positively correlated with climatic distance. GLM: Slope = 0.3552 (95% CI: 0.1083–0.6022); Y-intercept = 0.3011 (95% CI: 0.1975–0.4048); X-intercept = −0.8477 (95% CI: −3.679–−0.3333); R^2^ = 0.0339; F = 8.031; *p* = 0.005.

**Figure 4 plants-13-01334-f004:**
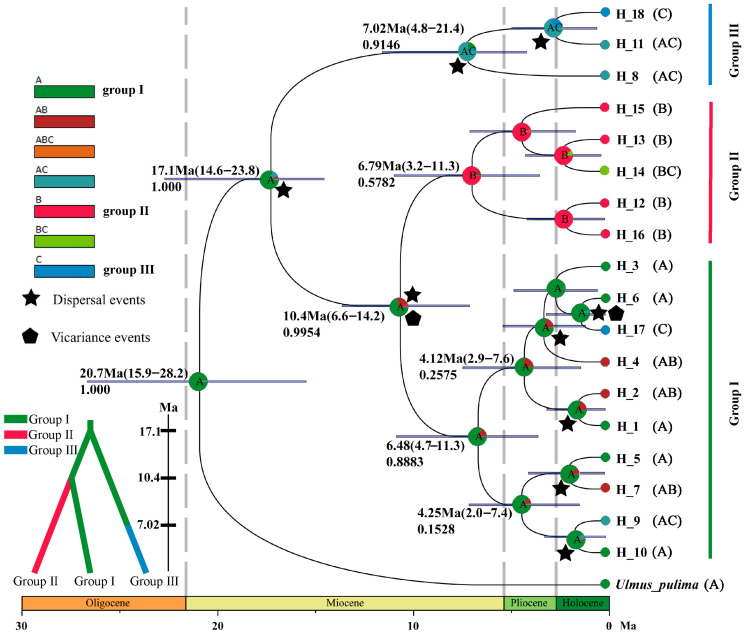
BEAST-derived phylogenetic relationships and ancestral distributions reconstruction by RASP analysis. On the BI tree, the numbers below nodes denoted a posterior probability, while the numbers above nodes indicated the divergence time with 95% highest posterior density (HPD).

**Figure 5 plants-13-01334-f005:**
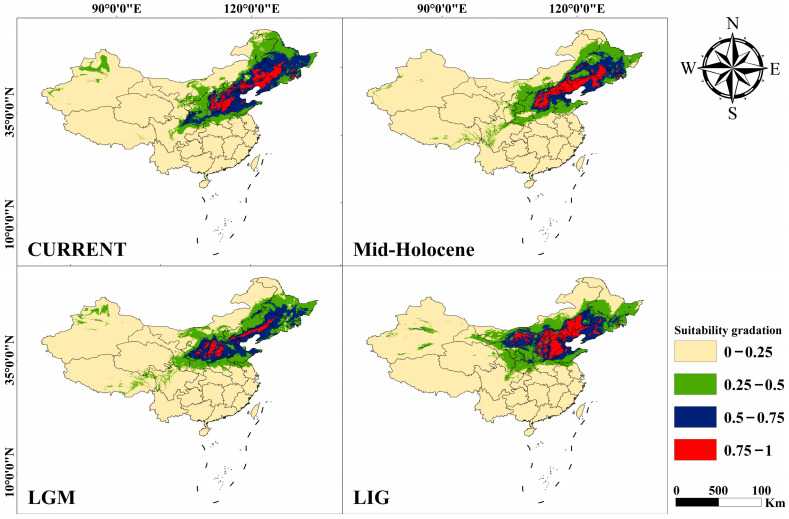
The estimated geographic distribution of *U. macrocarpa* in the current, Mid-Holocene, LGM, and LIG. Different colors correspond to different grades of suitability.

**Table 1 plants-13-01334-t001:** Landscape of genetic variation within and among three identified groups.

	Source	df	SS	MS	Est. Var.	Var. Per. (%)	F-Statistic (*p* < 0.05)
Among three groups	Among Groups	2	86.067	43.034	1.147	53%	*F*_CT_ = 0.532 *F*_SC_ = 0.141 *F*_ST_ = 0.598
	Among Pops	19	30.051	1.582	0.143	7%	
	Within Pops	88	76.400	0.868	0.868	40%	
	Total	109	192.518		2.158	100%	
Group Ⅰ vs. Group Ⅱ	Among Groups	1	32.995	32.995	0.797	44%	*F*_CT_ = 0.440 *F*_SC_ = 0.150 *F*_ST_ = 0.524
	Among Pops	14	22.717	1.623	0.152	8%	
	Within Pops	64	55.200	0.863	0.863	48%	
	Total	79	110.913		1.811	100%	
Group Ⅰ vs. Group Ⅲ	Among Groups	1	60.684	60.684	1.634	59%	*F*_CT_ = 0.587 *F*_SC_ = 0.155 *F*_ST_ = 0.651
	Among Pops	13	24.222	1.863	0.178	6%	
	Within Pops	60	58.400	0.973	0.973	35%	
	Total	74	143.307		2.785	100%	
Group Ⅱ vs. Group Ⅲ	Among Groups	1	35.023	35.023	1.047	55%	*F*_CT_ = 0.554 *F*_SC_ = 0.105 *F*_ST_ = 0.601
	Among Pops	11	13.162	1.197	0.089	5%	
	Within Pops	52	39.200	0.754	0.754	40%	
	Total	64	87.385		1.889	100%	
Total	Among Pops	21	116.118	5.529	0.932	52%	*F*_ST_ = 0.518
	Within Pops	88	76.400	0.868	0.868	48%	
	Total	109	192.518		1.80	100%	

*F*_CT_, genetic differentiation among groups; *F*_SC_, genetic differentiation among populations within groups; *F*_ST_, genetic differentiation among populations.

**Table 2 plants-13-01334-t002:** Rates of historical and contemporary gene flows per generation among three groups as calculated by the programs MIGRATE–N and BYESASS, respectively.

	Group Ⅰ	Group II	Group III
MIGRATE–N			
group Ⅰ	-	0.6451 (0.4540–2.1406)	1.3744 (0.3632–2.9279)
group Ⅱ	1.3054 (0.3393–2.7281)	-	0.6251 (0–2.0838)
group Ⅲ	0.9069 (0.186–2.0686)	0.8154 (0.1240–2.0686)	-
BAYESASS			
group Ⅰ	0.9511 (0.9178–0.9844)	0.0342 (0.0015–0.9844)	0.0403 (0.0038–0.0768)
group Ⅱ	0.0068 (0–0.0199)	0.9571 (0.9212–0.9930)	0.0201 (0–0.0464)
group Ⅲ	0.0420 (0.0108–0.0732)	0.0086 (0–0.0256)	0.9396 (0.8971–0.9821)

Values in parentheses represent 95% confidence intervals.

**Table 3 plants-13-01334-t003:** Degree of niche overlap based on Schoener’s *D* and Hellinger distance *I* statistics.

	Current	MH	LGM	LIG
Current	-	0.9647 ^+^	0.9500 ^+^	0.9306 ^+^
MH	0.8155 *	-	0.9563 ^+^	0.9265 ^+^
LGM	0.7900 *	0.7924 *	-	0.9327 ^+^
LIG	0.7218 *	0.7229 *	0.7341 *	-

The “*” represent the *D* value, and the “^+^” represent the *I* value.

## Data Availability

The datasets supporting the conclusions of this article are included within this article and its additional files. The datasets used and/or analyzed during the current study are available from the corresponding author upon reasonable request. The data are not publicly available due to its usage in current and confidentiality policy of the author’s laboratory.

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
