# Peer review of "The Phylogeography of Deciduous Tree Ulmus macrocarpa (Ulmaceae) in Northern China"

_plants, 2024, doi:10.3390/plants13101334_

Round 1

Reviewer 1 Report

Comments and Suggestions for Authors

The Phylogeography of Perennial Tree Ulmus macrocarpa (Ulmaceae) in Northern China.

The authors present a thorough phylogreographic study of the widespread tree species Ulmus macrocarpa  in northern China, a tree that is used as timber, but also has lots of ancillary medicinal uses. The study includes several key points, from the identification of haplotypes and analysis of genetic diversity to the phylogeographic inferences, but also includes an ecological niche modelling to predict the potential distribution of U. macrocarpa in China during different periods. The methods are all appropriate and relevant and the results are informative and well presented. In general, the manuscript is (very) well written with very few typos.

The manuscript

The few specific comments include:

Line 2: The title would be better without “perennial trees”. I suggest it to be simply “The Phylogeography of Ulmus macrocarpa (Ulmaceae) in Northern China”.

Line 33: “ a hotspot” does not really fit in this sentence. Could be replace by “central”

Line 60-62: “It has been reported that Ulmaceae emerged in the Early Cretaceous and could be separated into a temperate clade and a tropical clade”. Please add reference.

Line 70: Delete “The”. The Ulmus macrocarpa, mainly …

Line 84: “species” shouldn’t be italicized.

Line 84-85: Seed oil is often used in the food and industrial sectors and can also be used for brewing, making soy sauce, and for medicinal purposes.

Line 90: “delete “the” in As an important germplasm resource in Ulmaceae, although the U. macrocarpa had…”

Line 91 – 92: Consider changing “relatively scanty” in …. the advanced molecular phytogeography research on U. macrocarpa was relatively scanty compared to other Ulmaceae species.”

Line 245: add “the” in “In the structure analysis, the ΔK got the…”

Line 265: Consider deleting “were anxious” in  “Furthermore, we were anxious to investigated details about the genetic differentiation …”

Line 379: The physical geography in northern China is highly diversified. What does “physical geography” mean here? The topography?

Comments on the Quality of English Language

The English language is excellent. 

Author Response

Response to Reviewer 1 Comments

Dear reviewer,

Thank you for your valuable comments. We have studied the valuable comments from you carefully, and tried our best to revise our manuscript (Plants-2970994). Hope our revision could improve the manuscript. Thank you. The point to point responds to your comments as following:

Open Review

( ) I would not like to sign my review report

(x) I would like to sign my review report

Quality of English Language

( ) I am not qualified to assess the quality of English in this paper
( ) English very difficult to understand/incomprehensible
( ) Extensive editing of English language required
( ) Moderate editing of English language required
(x) Minor editing of English language required
( ) English language fine. No issues detected

Yes

Can be improved

Must be improved

Not applicable

Does the introduction provide sufficient background and include all relevant references?

(x)

( )

( )

( )

Are all the cited references relevant to the research?

(x)

( )

( )

( )

Is the research design appropriate?

(x)

( )

( )

( )

Are the methods adequately described?

(x)

( )

( )

( )

Are the results clearly presented?

(x)

( )

( )

( )

Are the conclusions supported by the results?

(x)

( )

( )

( )

Comments and Suggestions for Authors

Point 1: The authors present a thorough phylogreographic study of the widespread tree species Ulmus macrocarpa in northern China, a tree that is used as timber, but also has lots of ancillary medicinal uses. The study includes several key points, from the identification of haplotypes and analysis of genetic diversity to the phylogeographic inferences, but also includes an ecological niche modelling to predict the potential distribution of U. macrocarpa in China during different periods. The methods are all appropriate and relevant and the results are informative and well presented. In general, the manuscript is (very) well written with very few typos.

Response: Dear reviewer, thank you for your professional and positive comments and suggestions. Thank you for your efforts and time for reviewing our manuscript, sincerely.

Point 2: Line 2: The title would be better without “perennial trees”. I suggest it to be simply “The Phylogeography of Ulmus macrocarpa (Ulmaceae) in Northern China”.

Response: Dear reviewer, thank you for your suggestion. According to your suggestion and advice from reviewer 2, we deleted ‘Perennial’ and replace it by ‘Deciduous’. Thank you. Hope our revision could be satisfied with you.

Point 3: Line 33: “ a hotspot” does not really fit in this sentence. Could be replace by “central”.

Response: Dear reviewer, thank you for your suggestion. We have revised it.

Point 4: Line 60-62: “It has been reported that Ulmaceae emerged in the Early Cretaceous and could be separated into a temperate clade and a tropical clade”. Please add reference.

Response: Dear reviewer, we have added the references. Thank you.

Point 5: Line 70: Delete “The”. The Ulmus macrocarpa, mainly …

Response: Dear reviewer, we have deleted it. Thank you.

Point 6: Line 84: “species” shouldn’t be italicized.

Response: Dear reviewer, we have modified it.

Point 7: Line 84-85: Seed oil is often used in the food and industrial sectors and can also be used for brewing, making soy sauce, and for medicinal purposes.

Response: Dear reviewer, we have added the word “for” in this sentence. Thank you for your warm suggestion.

Point 8: Line 90: “delete “the” in As an important germplasm resource in Ulmaceae, although the U. macrocarpa had…”

Response: Dear reviewer, we have deleted it. Thank you for your suggestion.

Point 9: Line 91 – 92: Consider changing “relatively scanty” in …. the advanced molecular phytogeography research on U. macrocarpa was relatively scanty compared to other Ulmaceae species.”

Response: Dear reviewer, thank you for your suggestion. We have revised it.

Point 10: Line 245: add “the” in “In the structure analysis, the ΔK got the…

Response: Dear reviewer, thank you for your advice. We have added it in this sentence. Thank you.

Point 11: Line 265: Consider deleting “were anxious” in  “Furthermore, we were anxious to investigated details about the genetic differentiation …”

Response: Dear reviewer, we have deleted it. Thank you.

Point 12: Line 379: The physical geography in northern China is highly diversified. What does “physical geography” mean here? The topography?

Response: Dear editor, we have changed it into “topography” according to your advice, thank you.

Finally, Thank you for your consideration our manuscript of “The Phylogeography of Deciduous Tree Ulmus macrocarpa (Ulmaceae) in Northern China”. We would like to express our gratitude to the reviewer and editor for your professional comments and hardworking, which has significantly improved our manuscript. We hope that our revisions can meet the approving. Thank you again.

Sincerely,

Prof. Yiling Wang

Reviewer 2 Report

Comments and Suggestions for Authors

Plants, (plants-2970994-May 24)

Article: The Phytogeography of the Perennial Tree Ulmus macrocarpa

Authors: Ye H. et al.

Comments for the authors

General remark

This manuscript widens essentially our understanding about the demographic pattern of the important deciduous tree Ulmus macrocarpa today and during the glacial periods in Northern China. In addition, it is a very valuable contribution to our knowledge about refuges for trees during ice ages and their subsequent spreading within interglacial periods worldwide.

Therefore, it warrants publication.

In the opinion of this reviewer, the manuscript only needs a few formal improvements (see title) and a little shortage (compare ‘Detailed comments’).

Detailed comments

Headline

It is recommended to delete ‘Perennial’ and replace it by ‘Deciduous’. All trees are per se perennial. Annual trees are not known.

Abstract

Page 1

Line 18: Please avoid abbreviations like Nst and Gst in the ‘Abstract’ and prefer full names (only) at the beginning of the text!

Introduction

Page 2

Lines 69-79: A detailed description of the species U. macrocarpa is not necessary in this kind of study and can be read in books about the flora of China.

Lines 80-89: This can also be found somewhere else and must not be repeated in context with this study,

Material and Methods

This reviewer must admit that he is not familiar with all those methods and models described in this chapter. Nevertheless, he is impressed by the great number of efforts and work details.

Results

Page 7

Lines 258 etc.: In most international journals the lower-case letter p is used instead of P (also in lines 274, 275, and in Table 1).

Line 266:  AMOVA or ANOVA?

Page 9

Figure 3 and the text: Authors could think about the problem: Is it realistic to use so many decimal places behind the point of all the number (also in Table 1)?

Page 10

Line 313:  What means ‘sates’ here (states?)?

Page 11

Table 2: Too many digits behind the point of all the numbers!

Page 12

Table 3: Are all these numbers behind the point necessary.

Discussion

No comments

References

Citations were only checked at random by this reviewer. It is recommended that the authors control them all again.

Line 518: Scientific species names should be in italics!

Page 16

Lines 533, 536, 539, etc. … 574, 577: Scientific names should be in italics!

Page 17

Lines 580, 594: Italics.

Page 18

Lines 674,685, … 696: Italics.

Page 19

Lines 730, 739: Italics.

Final comment

This manuscript should be published after minor corrections.

Author Response

Response to Reviewer 2 Comments

Dear reviewer,

Thank you for your valuable comments. We have studied the valuable comments from you carefully, and tried our best to revise our manuscript entitled “The Phylogeography of Deciduous Tree Ulmus macrocarpa (Ulmaceae) in Northern China” (Plants-2970994). Hope our revision could improve the manuscript. Thank you. The point to point responds to your comments as following:

Open Review

Quality of English Language

(x) I am not qualified to assess the quality of English in this paper

( ) English very difficult to understand/incomprehensible

( ) Extensive editing of English language required

( ) Moderate editing of English language required

( ) Minor editing of English language required

( ) English language fine. No issues detected

Yes

Can be improved

Must be improved

Not applicable

Does the introduction provide sufficient background and include all relevant references?

(x)

( )

( )

( )

Are all the cited references relevant to the research?

( )

(x)

( )

( )

Is the research design appropriate?

(x)

( )

( )

( )

Are the methods adequately described?

(x)

( )

( )

( )

Are the results clearly presented?

(x)

( )

( )

( )

Are the conclusions supported by the results?

( )

( )

( )

( )

Comments and Suggestions for Authors

Point 1: This manuscript widens essentially our understanding about the demographic pattern of the important deciduous tree Ulmus macrocarpa today and during the glacial periods in Northern China. In addition, it is a very valuable contribution to our knowledge about refuges for trees during ice ages and their subsequent spreading within interglacial periods worldwide.

Therefore, it warrants publication. In the opinion of this reviewer, the manuscript only needs a few formal improvements (see title) and a little shortage

Response: Dear reviewer, thank you for your effort for reviewing our manuscript and we are gratefully of your positive comments. Thank you, sincerely.

Point 2: It is recommended to delete ‘Perennial’ and replace it by ‘Deciduous’. All trees are per se perennial. Annual trees are not known.

Response: Dear reviewer, thank you for your suggestions. We have revised it.

Point 3: Line 18: Please avoid abbreviations like Nst and Gst in the ‘Abstract’ and prefer full names (only) at the beginning of the text.

Response: Dear reviewer, Thank you for your suggestions. The GST and NST are two measures of differentiation can be compared for a single data set: the GST that makes use only of the allelic (haplotype) frequencies while the NST for which similarities between the haplotypes are taken into account in addition. Indeed, they were both the parameters for assessing the level of genetic differentiation. We revised the abstract according your suggestions. And we also modified the relevant contents (Line 148 to Line 150) in Method to make it more clear. Thank you for your suggestions.

Point 4: Lines 69-79: A detailed description of the species U. macrocarpa is not necessary in this kind of study and can be read in books about the flora of China.

Response: Dear reviewer, we have reduced this paragraph to make it more brief. Thank you for your advice.

Point 5: Lines 80-89: This can also be found somewhere else and must not be repeated in context with this study

Response: Dear reviewer, thank you for your suggestions. We have shorted this paragraph to make it brief.

Point 6: This reviewer must admit that he is not familiar with all those methods and models described in this chapter. Nevertheless, he is impressed by the great number of efforts and work details.

Response: Dear reviewer, thank you for your positive comments.

Point 7: Lines 258 etc.: In most international journals the lower-case letter p is used instead of P (also in lines 274, 275, and in Table 1).

Response: Dear reviewer, thank you for your suggestions. We checked the manuscript and changed them into lower-case. Thank you for your kind remind.

Point 8: Line 266: AMOVA or ANOVA?

Response: Dear reviewer, AMOVA is correct here, it means “analysis of molecular variance”.

Point 9: Figure 3 and the text: Authors could think about the problem: Is it realistic to use so many decimal places behind the point of all the number (also in Table 1)?

Response: Dear reviewer, thank you for your suggestions. Different analyses yield results with varying degrees of accuracy, reflected by different decimal digit. We revised the Figure 3 and the text. We modified the decimal places no more than four in all Tables and the text. Thank you for your suggestions.

Point 10: Line 313: What means ‘sates’ here (states?)?

Response: Dear reviewer, thank you for your suggestions. We have changed the “sates” into “states”. Sorry for our misspelling.

Point 11: Table 2: Too many digits behind the point of all the numbers.

Response: Dear reviewer, thanks for your suggestion. We have revised the it. Thank you.

Point 12: Table 3: Are all these numbers behind the point necessary.

Response: Thanks for your advice. According to your suggestion, we keep the decimal places up to four in our manuscript, to ensure both the standardization of academic writing and the accuracy of analytical results.

Point 13: Citations were only checked at random by this reviewer. It is recommended that the authors control them all again.

Response: Dear reviewer, thanks for your concerns. We have checked the references in this manuscript again. Thank you.

Point 14: Line 518: Scientific species names should be in italics!

Lines 533, 536, 539, etc. … 574, 577: Scientific names should be in italics!

Lines 580, 594: Italics.

Lines 674,685, … 696: Italics.

Lines 730, 739: Italics.

Response: Dear reviewer. We have revised them and check the whole Reference section. Thank you very much.

Finally, Thank you for your consideration our manuscript of “The Phylogeography of Deciduous Tree Ulmus macrocarpa (Ulmaceae) in Northern China”. We would like to express our gratitude to the reviewer and editors for your professional comments and hardworking, which has significantly improved our manuscript. We hope that our revisions can meet the approving. Thank you again.

Sincerely,

Prof. Yiling Wang
